# Molecular Characterization of *Listeria monocytogenes* in the Food Chain of the Republic of Kosovo from 2016 to 2022

**DOI:** 10.3390/foods13182883

**Published:** 2024-09-12

**Authors:** Besart Jashari, Karine Capitaine, Bledar Bisha, Beatrix Stessl, Katerina Blagoevska, Armend Cana, Dean Jankuloski, Benjamin Félix

**Affiliations:** 1Food Microbiology, Food and Veterinary Laboratory, Food and Veterinary Agency of Kosovo, Lidhja e Pejës 241, 10000 Pristina, Kosovo; besart.jashari@rks-gov.net; 2Laboratory for Food Safety, Salmonella and Listeria Unit, ANSES, European Union Reference Laboratory for *L. monocytogenes*, University of Paris-Est, 94700 Maisons-Alfort, France; karine.capitaine@anses.fr (K.C.); benjamin.felix@anses.fr (B.F.); 3Department of Animal Science, College of Agriculture and Natural Resources, University of Wyoming, Laramie, WY 82071, USA; 4Unit of Food Microbiology, Centre for Food Science and Public Veterinary Health, Clinical Department for Farm Animals and Food Systems Safety, University of Veterinary Medicine Vienna, Veterinärplatz 1, A-2110 Vienna, Austria; beatrix.stessl@vetmeduni.ac.at; 5Food Institute, Faculty of Veterinary Medicine—Skopje, Ss. Cyril and Methodius University in Skopje, Lazar Pop-Trajkov 5–7, 1000 Skopje, North Macedonia; katerinab@fvm.ukim.edu.mk (K.B.); djankuloski@fvm.ukim.edu.mk (D.J.); 6Microbiology Laboratory, University for Business and Technology—Higher Education Institution, Kalabria, 10000 Pristina, Kosovo; armend.cana@ubt-uni.net

**Keywords:** *L. monocytogenes*, food chain, phylogenetic lineage, molecular serotype, clonal complex

## Abstract

The present study describes the genetic characterization of *L. monocytogenes* strains found in the Republic of Kosovo’s food chain. From 2016 to 2022, 995 samples were collected. Overall, 648 samples were from ready-to-eat (RTE) food products, 281 from food products consumed cooked (FPCC), 60 from raw materials, and 6 from environmental samples. Overall, 11.76% (117 out of 995) of the samples were contaminated by *L. monocytogenes*, comprising 6.33% (41 out of 648) from RTE products, 14.95% (42 out of 281) from FPCC, 55.00% (33 out of 60) from raw materials, and 16.66% (1 out of 6) from environmental samples. All isolates were subjected to molecular serotyping and clonal complex (CC) identification by using real-time PCR, as well as multilocus sequence typing. All isolates were grouped into four molecular serotypes, IIa (34.19%), IIb (3.48%), IIc (32.48%), and IVb (29.91%), as well as Lineage I (33.33%) and Lineage II (66.66%). In total, 14 CCs were identified from 41 RTE isolates; however, CC29 (7), CC2 (6), and CC6 (6) were the most dominant. By contrast, CC9 was by far the most represented CC in both FPCC (21) and RM (14). Moreover, 30 isolates expressed CC1, CC2, CC4, or CC6, which are particularly associated with severe human infections.

## 1. Introduction

*Listeria monocytogenes* (*L. monocytogenes*) is a Gram-positive, non-spore-forming facultative anaerobic bacterium. It is considered to be ubiquitous and capable of surviving or thriving in a variety of foodstuffs, and it can grow at refrigeration temperatures, survive freezing conditions and high salt concentrations, and tolerate the presence of nitrites [1,2]. *L. monocytogenes* is considered to be widespread in different environments, including water, dust, and soil [3]. This bacterium is recognized as a major foodborne pathogen and causes listeriosis, an illness that often occurs in the form of sporadic infections and is associated with high mortality rates in susceptible individuals [4].

Listeriosis is commonly associated with the consumption of ready-to-eat (RTE) foods, such as minimally processed foods, soft cheese, and especially RTE meat products [5]. While the incidence of listeriosis is low, it is a severe disease, with a reported case fatality rate of 18.1% in 2022 in Europe [6] and similar mortality reported in the United States over the past decade (15.3%) [7]. One major reason for listeriosis outbreaks often reported in recent investigations [8] is the persistence of *L. monocytogenes* in food processing environments [9]. Listeriosis is a burden to society, public health, and the economy [10,11] and requires constant molecular surveillance to link human cases to food vehicles and their sources of contamination.

Thirty years ago, the consumption of soft cheese was linked to 49% of sporadic instances of listeriosis in France, but preventive initiatives have decreased the frequency of sporadic cases of listeriosis there by 68% from 1987 to 1997 [12]. These results were made possible by strong molecular surveillance linking food and human cases [13] and the implementation of microbiological criteria at the European Union (EU) level, strictly applied nationally [14].

The Republic of Kosovo (RKS) is a small-sized, newly established country in the Balkans, with small-scale farming and food processing facilities and in need of economic growth to satisfy the needs of its population, especially in the sector of animal production. As animal production increases, this will need to be met with a robust surveillance system for foodborne pathogens, including molecular surveillance to mirror requirements in the EU countries. Prior to this study, there was a paucity of information regarding molecular typing data on bacterial isolates circulating in Kosovo, including those that were foodborne and from clinical cases of listeriosis. Limited data have been published on the prevalence of *L. monocytogenes* in North Macedonia [15] in Serbia [16].

For countries surrounding the RKS, the prevalence of *L. monocytogenes* in RTE meat products of beef origin was between 2.7 and 3.9%, and for hard cheeses, it was 4.6% [6]. Listeriosis cases from animals and levels of contamination in foods have already been reported in Kosovo from ovine clinical cases [17], on-farm milk products [18], retail cheeses [19], and the meat sector [20]. However, these studies did not provide the molecular characterization of the strains and were conducted in a limited number of retail sites, processing plants, or farms. Consequently, although these studies provide useful information on the occurrence of listeriosis in the animal sector and provide baseline data on levels of contamination with *L. monocytogenes* in foods for the country, their utility is limited in terms of context with other studies conducted regionally or at the global level.

*L. monocytogenes* is divided into four lineages. Lineages I and II are the most widespread, frequently isolated from food, natural environments, and farms, as well as cases of sporadic animals or humans. Lineages I and II are also found more frequently in human outbreaks, and over 95% of listeriosis disease cases are linked to these lineages [21,22].

The *L. monocytogenes* species is divided into four molecular serotypes—IIa, IIc (Lineage II), IIb, and IVb (Lineage I) [23]—as determined via the traditional, conventional serotyping methods. Since 2008, molecular serotyping has been supplemented by multilocus sequence typing (MLST). This method provides a universal language for strain typing [24,25]. MLST classifies *L. monocytogenes* according to sequence type (ST) and clonal complex (CC). STs are defined as the unique association of alleles from seven housekeeping genes, and a CC is described as a cluster of STs sharing at least six alleles. Strains from the same CCs descend from a common ancestor and have accumulated differences, predominantly through mutations. Today, with access to advanced technology such as whole-genome sequencing (WGS) and the assessment of the core genome multilocus sequence typing (cgMLST) from the data derived from WGS, it is possible to type and discriminate the different strains of an outbreak with high accuracy, thus tracing the source of listeriosis [13,26,27]. However, not all laboratories in developing countries have access to WGS services in a reasonable time during outbreaks. Therefore, an approach targeting CCs by high-throughput real-time PCR, which enables the rapid identification of the 30 major MLST CCs circulating in Europe, is particularly helpful for strain screening prior to WGS [28].

The present study aims to characterize the diversity of *L. monocytogenes* strains in terms of lineage, molecular serotype, and CCs using this multiplex PCR method, in order to provide an overview of the genetic diversity of the strains circulating among food products in Kosovo. This knowledge can then be used to better understand human listeriosis outbreaks in the country and inform public health measures and outbreak investigation efforts.

## 2. Materials and Methods

### 2.1. Sampling

A total of 995 samples were processed, with five units per sample, resulting in a total of 4975 tests. The analyzed samples were taken from RTE foods, food products consumed cooked, raw materials, and environmental samples. The food products sampled were sorted into one of two categories, with one category including foods that supported the growth of *L. monocytogenes* and the other category comprising foods that did not support its growth [14].

The samples were analyzed for the presence of *L. monocytogenes* from January 2016 to March 2022. The tested samples of animal origin were collected in the RKS at all stages of food chain processing. Out of these samples, 648 samples were from RTE foods (meat products n = 286, milk products n = 361, and fish products n = 1), 281 were from food products consumed cooked (meat products n = 249 and milk products n = 32), 60 were from raw materials (meat n = 39, milk n = 8, fish n = 8, and combined food products n = 5), and six were from various environments (food contact samples n = 5 and sample from personnel n = 1). Most were typical local products reflecting the taste of Kosovar gastronomic traditions and artisanal food culture. The samples were obtained according to the food code under aseptic conditions. Each sample was marked and registered for identification purposes prior to storage under optimal storage conditions (+5 °C). Based on the Regulation on Microbiological Criteria 2073:2005, a representative sample consisted of five units of 25 g for each tested sample. *L. monocytogenes* isolates were detected in the collected samples via the official EN ISO 11290-1 method with two-phase enrichment [29].

### 2.2. Detection and Isolation

In accordance with Mandate M381 by the European Commission for CEN, inter-laboratory comparisons validated the reference technique for the detection and reporting of *L. monocytogenes* in food (standard ISO 11290-1). The specificity and sensitivity of the detection method in these food matrices varied from 97.6% to 100% and from 91.1% to 100%, respectively [30]. From each sample, 25 g was extracted and inoculated into 225 mL of Demi-Fraser Broth (DFB, Liofilchem^®^ S.r.l, Roseto degli Abruzzi, TE, Italy) for initial selective enrichment. After incubation at 30 °C ± 1 °C for 24 h ± 2 h, 0.1 mL of the liquid culture was inoculated into 10 mL of full-strength Fraser Broth (FB, Liofilchem^®^ S.r.l, Roseto degli Abruzzi, TE, Italy) for the second enrichment and cultured at 37 °C ± 1 °C for 24 h ± 2 h. Listeria Ottaviani and Agosti Agar (ALOA, Liofilchem^®^ S.r.l, Roseto degli Abruzzi, TE, Italy) and Oxford Listeria Selective Agar (LOA, Liofilchem^®^ S.r.l, Roseto degli Abruzzi, TE, Italy) were used as selective and differential media, in Petri dishes incubated at 37 °C ± 1 °C for 24 h to 48 h.

### 2.3. Biochemical Identification

Five typical and five atypical colonies were subcultured onto tryptic soy agar supplemented with 0.6% yeast extract (TSA-YE, Liofilchem^®^ S.r.l, Roseto degli Abruzzi, TE, Italy) as a non-selective medium and incubated at 37 °C for 24 h.

The isolates were confirmed as *L. monocytogenes* via Gram staining, catalase reaction, oxidase testing, carbohydrate utilization, hemolysis testing, CAMP testing, and motility at 20 °C to 25 °C. *L. monocytogenes* was confirmed and phenotypically identified to the species level using standard API Listeria biochemical tests (Bio Mérieux, Marcy l’Etoile, France) [31], following the manufacturer’s instructions. *L. monocytogenes* ATCC (American Type Culture Collection) 13932 and 35152 (Liofilchem^®^ S.r.l, Roseto degli Abruzzi TE, Italy), were utilized as control strains.

### 2.4. DNA Extraction

DNA was extracted from pure isolates using the specific kit for Gram-positive bacteria—Mericon DNA Bacteria Plus Kit (Qiagen, Hilden, Germany). Pure bacterial cultures were transferred into brain heart infusion broth (BHIB, Liofilchem^®^ S.r.l, Roseto degli Abruzzi, TE, Italy) and incubated for 24 h at 37 °C ± 1 °C. One milliliter of supernatant was boiled in a water bath at 100 °C for 10 min and then centrifuged at 13,000× *g* for 5 min (Eppendorf 5415R refrigerated centrifuge, USA). Next, 400 µL of rapid lysis buffer was added to the bacterial pellet, vortexed vigorously, and then pelleted 13,000× *g* for 5 min. The supernatant was kept at −20 °C for a maximum of three weeks.

### 2.5. DNA Concentration

The DNA concentration was measured using the Qubit™ dsDNA HS with a Qubit^®^ 2.0 fluorometer (Thermo Fisher Scientific, Waltham, MA, USA). A DNA extract was retained when its concentration was above 0.1 ng/µL.

### 2.6. Detection of Lineages, Molecular Serogroups, and Clonal Complexes

The concentration of DNA extracts was adjusted to between 0.1 and 1 ng/µL; then, real-time multiplex PCR was performed using Taqman^®^ PCR probes from TIB Molbio, (Berlin, Germany), according to the methodology described by Félix et al. 2023 [28]. Three different real-time PCR thermocyclers were used to perform the tests: a Mic-4 real-time PCR thermocycler from Bio Molecular Systems (Upper Coomera, Australia), a Rotor-Gene Q real-time PCR thermocycler from QIAGEN (Hilden, Germany), and a QuantStudio 5 real-time PCR thermocycler from Thermo Fisher Scientific (Waltham, MA, USA).

The following target genes were used for molecular serotyping [23]: *lmo0737*, *lmo1118*, *ORF2819*, *ORF2110*, *prs*, and *plca*. Clonal complexes (CCs) were identified by amplifying CC-specific genomic regions [28]: CC1, CC2, CC3, CC4, CC5, CC6, CC7, CC8, CC9, CC11-ST451, CC14-ST14-206-399, CC18, CC19-ST398-802-1308, CC20, CC21, CC26, CC29, CC31, CC37, CC54, CC59, CC77, CC87, CC101, CC121, CC155, CC193, CC199, CC204, and CC224. Positive controls were provided by the European Reference Laboratory for *L. monocytogenes*. Positive amplifications were determined when reactions with a detection threshold less than or equal to 30 cycles (Ct ≤ 30) were recorded, while late reactions, over 30 cycles (Ct > 30), were considered to be non-specific. Additionally, multilocus sequence typing (MLST) was used to complement the results with ST32, which was not covered by the CC real-time PCR panel. The CC strain obtained was displayed within a minimum spanning tree, built using BioNumerics 7.6.3 (BioMérieux Applied Maths, Sint-Martens-Latem, Belgium), with default parameters.

## 3. Results

### 3.1. L. monocytogenes in Food Products

A total of 995 samples were tested, of which 117 were determined to be positive for *L. monocytogenes*. The strains of *L. monocytogenes* were confirmed via biochemical and molecular methods, as stated in the Methods section. The 117 strains were routinely isolated from the four major food sectors and four food categories throughout Kosovo over six years (Table 1 and Figure 1).

Most of the isolates were obtained from meat and milk products from RTE, FPCC, and raw materials. The prevalence in meat products and dairy products can be taken as a statistical value in this study considering the large number of examined samples. The overall prevalence of *L. monocytogenes* in the two major food categories sampled in this study was 14.29% and 6.39% for meat products and milk products (mostly cheese), respectively. The lowest prevalence was found in RTE foods (6.33%) compared to food products consumed cooked (14.95%) and raw materials (55.00%).

Within the category of RTE foods, the prevalence was lower in dairy products, at 5.82%, compared to meat products, at 6.99%. Fish and fishery products, as well as combined food products (meat and dairy), had a prevalence of 44.44% and 100.00%, respectively. Despite the limited number of samples that were available for these two categories, high prevalence was expected, similar to raw materials, which are prone to higher contamination rates, with an overall of 55.00%.

### 3.2. Genetic Diversity of L. monocytogenes

Among the 117 isolates of *L. monocytogenes* tested by means of multiplex PCR, we identified a total of 17 clonal complexes belonging to four molecular serotypes: IIa (34.19%), IIb (3.48%), IIc (32.48%), and IVb (29.91%). Further, they were divided into two phylogenetic lineages. Lineage I included 39 (33.33%) and Lineage II included 78 (66.66%) of the 117 isolates; lineages III and IV were not identified in any isolates (Figure 3).

#### 3.2.1. Lineages

Considering the food chain categories, Lineage I was identified in 48.78% of RTE, 23.81% of FPCC, and 27.27% of raw material isolates, whereas Lineage II was detected in 51.22% of RTE, 76.19% of FPCC, and 72.73% of raw material isolates (Table 2).

When divided by food category, out of the 82 isolates from meat products only, 29.27% belonged to Lineage I and 70.73% to Lineage II. By contrast, out of the 25 isolates from milk and milk products alone, 44.00% belonged to Lineage I and 56.00% to Lineage II. Moreover, four out of the five combined food isolates belonged to Lineage II and one to Lineage I. Three out of the four isolates from fish products belonged to Lineage I and one to Lineage II. There was only one isolate of *L. monocytogenes* from the environmental samples, which belonged to Lineage II (Table 2).

#### 3.2.2. Molecular Serotypes

Among the 41 isolates from RTE foods, the identified molecular serotypes were 18 (43.90%) each of IIa and IVb, 3 IIc (7.32%), and 2 IIb (4.88%). Of the 42 FPCC isolates, 21 (50.00%) belonged to IIc, followed by 11 (26.19%) to IIa, 9 (21.43) to IVb, and 1 (2.38%) to IIb molecular serotypes. Of the 33 raw material isolates, 14 (42.42%) belonged to IIc, followed by 10 (30.30%) to IIa, 8 (24.24%) to IVb, and 1 (3.03%) to IIb molecular serotypes.

When divided by food category, out of the 82 isolates from meat products only, IIc was the most represented molecular serotype, comprising 52 or 42.68% of the isolates, followed by IIa, with 23 samples (28.05%); IVb, with 22 samples (22.26%); and IIb, with 2 samples (2.44%). Out of the 25 isolates from milk and milk products alone, IIa was the most represented molecular serotype, comprising 12 (48.00%) of the isolates, followed by IVb, with 10 samples (40.00%); IIc, with 2 samples (8.00%); and IIb, with 1 (4.00%) isolate.

Moreover, out of the five isolates from combined food products, three (60.00%) belonged to the IIa molecular serotype, followed by one (20.00%) to IIc and one (20.00%) to IVb. Two out of the four isolates (50.00%) from fish products belonged to molecular serotypes IVb, IIa and IIb, with one sample each. The isolate from the environmental samples belonged to molecular serotype IIa (Table 2).

#### 3.2.3. Clonal Complexes

Notably, 14 out of the 17 clonal complexes were identified by real-time PCR from 41 RTE isolates. However, CC29, CC2, and CC6 were the most dominant, with seven CC29, followed by six samples in each of the CC2 and CC6 clonal complexes. By contrast, CC9 by far was the most detected clonal complex in both FPCC (21) and RM (14). Thirteen groups of clonal complexes were identified from meat product-only isolates. In total, 35 out of the 82 meat product isolates expressed CC9, which was the most represented clonal complex in this group, followed by CC6, which was expressed by 9 isolates; CC8 by 7 isolates; and CC2, expressed by 5 isolates. Notably, 11 groups of clonal complexes were identified from the 25 samples from milk products only. CC29 was identified in six isolates, followed by CC2 in five, while CC9 was only detected in two isolates. Three of the four isolates from fish products belonged to CC2, with two samples, and CC26 and CC87 with only one each. From the four groups of clonal complexes isolated from combined food products, we found that CC121, which we did not detect in any other food category, was expressed by two isolates, whereas CC6, CC8, and CC9 were expressed by one isolate each (Table 2 and Figure 2).

A minimum spanning tree (MST) was reconstructed based on food chain categories to determine the population structure of clonal complexes of *L. monocytogenes* in the Republic of Kosovo (Figure 3a,b).

## 4. Discussion

This is the first study of this kind carried out in the RKS. The study is novel because it analyzed large numbers and types of samples over extended periods of time and provided the first instance of the molecular characterization of *L. monocytogenes* from a variety of matrices in the country.

The sampling targeted mainly RTE meat and milk products or food products intended for consumption as cooked. *L. monocytogenes* was isolated from 117 out of a total of 995 samples examined (11.76%).

The highest contamination rates were found in raw materials (55.00%) and FPCC (14.95%), whereas RTE foods had the lowest prevalence (6.33%). This distribution highlights the risk associated with raw and minimally processed food products, which are more susceptible to contamination due to factors such as handling and storage conditions.

Within RTE products, the prevalence of *L. monocytogenes* was 5.82% for milk products (mostly cheese) and 6.99% for meat products. These figures are higher than the average prevalence reported in EU countries for both product categories, which stand at 0.37% and 2.1%, respectively [6]. Previous studies conducted in the Republic of Kosovo on RTE products reported a higher prevalence, with 6.9% for cheese [19] and 10.1% for meat products [20].

For meat products consumed cooked, the prevalence was 16.95%, higher than in a similar study conducted on raw sausages and raw meat in France, where the prevalence was 10% and 12%, respectively [32,33].

Due to the relatively small number of combined food products (meat and dairy) and fish product samples examined in this study, the calculated prevalence of *L. monocytogenes* cannot be taken as an accurate statistic, so it was not possible to assess the significance of the different distributions of serotypes between environmental and food strains.

The genetic diversity of the strains observed in Kosovo was compared with studies conducted either at the European level [27] or at the national level, in France [34,35], Switzerland [36], and Slovakia [37].

Only Lineage I (33.33%) and Lineage II (66.66%) strains were identified in the present study. Their lineage distribution ratio was consistent with the abovementioned studies, which found between 22 and 37% for Lineage I and between 69 and 78% for Lineage II.

The results of molecular serotyping provided a pre-screening analysis prior to CC identification. Molecular serotyping results enable direct comparisons with previous studies using only conventional serotyping or molecular serotyping as the typing method. Both methods provide a corresponding nomenclature. Briefly, with few exceptions, the major serotype correspondence was 1/2a = IIa, 1/2b = IIb, 1/2c = IIc and 4b = IVb. In the present study, four main molecular serotypes were identified: IIa (34.19%), IIb (3.48%), IIc (32.48%), and IVb (29.91%). The predominance of molecular serotypes IIa and IIc, accounting for over two-thirds of the isolates, aligns with previous studies reporting these molecular serotypes as prevalent in various geographical regions [38,39,40]. The diversity of molecular serotypes is indicative of the various sources and potential reservoirs of contamination within the food chain. Regarding the IIc (or 1/2c) strains, they have largely been reported in meat products in previous studies [41,42,43]. Conversely, they are rare in milk products [44,45]. These results show the large distribution of the IIc (or 1/2c) strain in European countries, including the RKS, during a long period of time. Regarding the IVb (or 4b) strain, this molecular serotype includes the CCs associated with severe human infections [35] (CC1, CC2, CC4, and CC6) and is considered hypervirulent. Overall, 26.83% of the meat samples and 40.00% of the milk product samples belonged to this molecular serotype.

Overall, 14 CCs were identified among the 41 RTE isolates, with CC29, CC2, and CC6 being the most dominant. Conversely, CC9 emerged as the most prevalent in FPCC and raw materials. This clonal complex is known for its presence in various environmental and food sources but is less commonly associated with severe human cases [46]. The high prevalence of CC9 in raw materials could suggest that it is a persistent environmental strain, possibly contributing to cross-contamination during food processing.

The most frequent CCs in meat products only were CC9, CC8, CC6, and CC7, whereas CC2 and CC29 were more common in milk products. In all the above studies, CC9 and CC8 were the most frequent CCs for meat, and although CC6 and CC7 were observed, they were not predominant. Surprisingly, CC121 was absent from meat products in the present study, although it was frequently associated with meat products elsewhere [35]. The absence of this clone in meat products in the current study needs further investigation but may reflect a difference in animal slaughtering or processing practices in Kosovo. For milk products, CC2 was the second most frequent CC in [27]. For the other studies, CC2 was observed in milk products but was not predominant. The CC29 clone was observed in milk products in all the studies above but was not predominant and less abundant than CC2. All 30 isolates expressing the CCs associated with severe human infections (CC1, CC2, CC4, or CC6) belonged to the molecular serotype IVb. This highlights the need to identify and confirm the presence of these CCs and map their distribution in the current food chain in order to predict the public health risks associated with food products.

Whole-genome sequencing (WGS) is recommended to be used in the future in order to provide additional information on virulence factors and other genetic determinants. In the context of the dynamic development of the Kosovo agri-food sector, *L. monocytogenes* typing will become a crucial tool for food safety management. The database created in this study made it possible to investigate outbreaks based on CC frontline screening, drawing links between human clinical cases and food contaminants. This step is essential for selecting strains before confirming outbreaks [13] with a sequence-based method such as core genome MLST (cgMLST), which is currently the reference method for molecular surveillance [6,47]. This is the method that would be applied in the RKS in the event of an outbreak through EU-supportive actions, potentially through the EURL’s program.

Without a doubt, researchers in the RKS need to optimize molecular methods using advanced technology and perform tests in real time in order to achieve effective surveillance. The remaining challenge is to further develop the surveillance and information system in Kosovo in both the short and medium term.

## 5. Conclusions

As a conclusion of this study covering the period 2016–2022, which included samples from the Republic of Kosovo, we obtained the results related to the general condition of *L. monocytogenes*. We found that contamination in RTE foods and meat products consumed as cooked is higher than the prevalence previously reported in the EU. Major hypervirulent clones CC1, CC2, CC4, and CC6 were identified in food products. These clones should be targeted as a priority using more rigorous control measures, especially to protect at-risk groups from this pathogen (the elderly population, pregnant women, children, and immunosuppressed persons). The effective management of the risk of *L. monocytogenes* in the food chain and food products requires a thorough assessment of sources of contamination, continuous monitoring, and rigorous control measures. By conducting detailed risk assessments and adapting strategies based on ongoing assessments, it is possible to reduce the risk of listeriosis and protect public health. It is necessary to continue the active monitoring of the entire food chain to support the real-time molecular surveillance of CCs present, which should also occur as soon as possible to investigate the identified CCs, at least retrospectively, with WGS and cgMLST.

## Figures and Tables

**Figure 1 foods-13-02883-f001:**
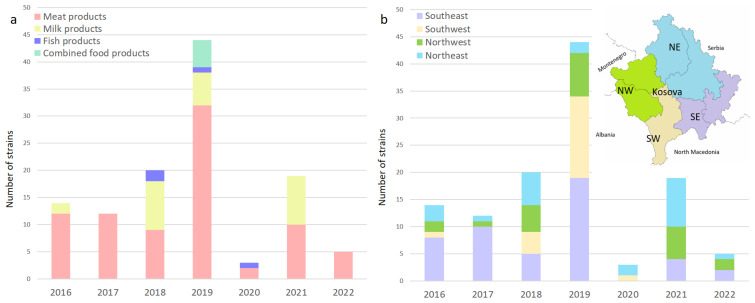
(**a**) Distribution of *L. monocytogenes* over the years in meat products, milk products, fish products, and combined products; (**b**) map showing the geographic location of sampling sites. Regional location of the sampling sites among business operators in Kosovo, with purple bars in the southeast (SE), brown in the southwest (SW), green in the northwest (NW), and blue in the northeast (NE).

**Figure 2 foods-13-02883-f002:**
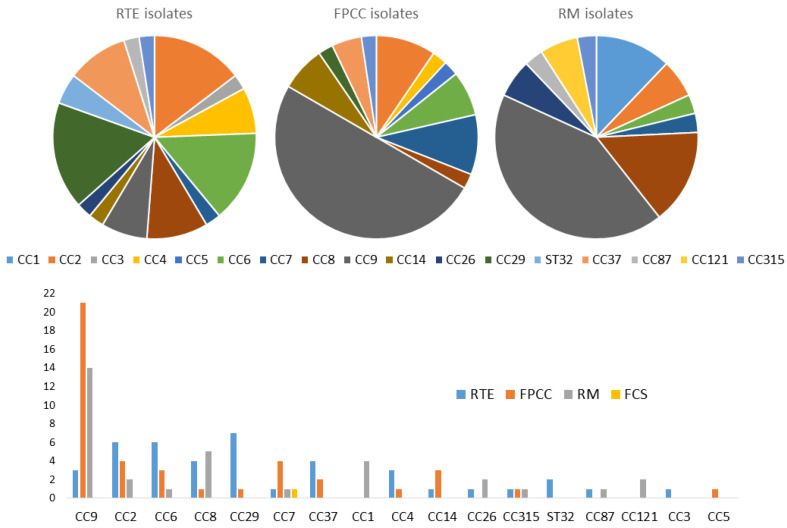
Graphical distribution of *L. monocytogenes* clonal complexes in the food chain.

**Figure 3 foods-13-02883-f003:**
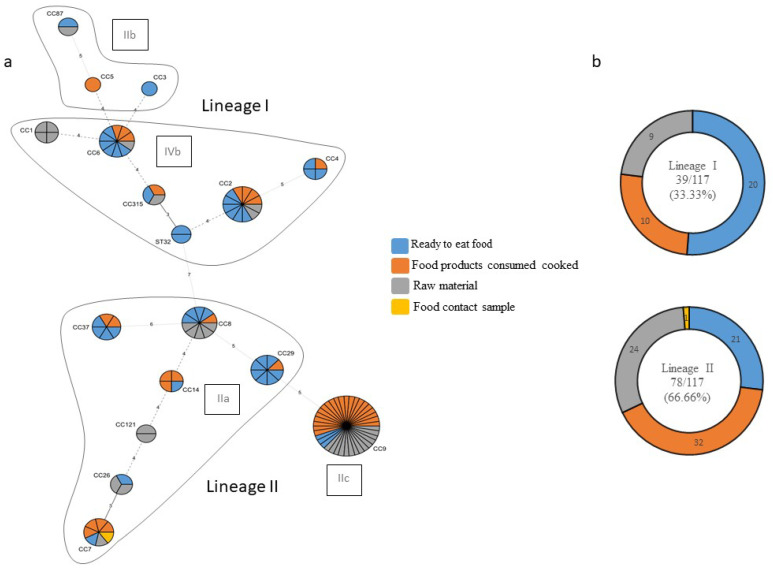
(**a**) The minimum spanning tree (MST) of multilocus sequence typing (MLST) clonal complexes (CCs) of the 117 *L. monocytogenes* strains included in the study panel. Each CC is indicated by a circular node whose size reflects the number of strains. The CCs used to build the MST were obtained by an MLST alternative method, which provides only the CC, to build the MST; the smallest ST allelic code within the CC was used. The numbers along the node connecting the lines indicate the number of allelic differences between them. The color reflects the food chain category: ready to eat food in blue, food products consumed cooked in orange, raw material in grey, and food contact sample in yellow. Each delimited area groups the CCs belonging to the same molecular serotype, indicated in a black frame; (**b**) strain distribution according to the lineage and type of food sector.

**Table 1 foods-13-02883-t001:** The number of samples per food category and the number of samples positive for *L. monocytogenes* in food chain and environmental samples in Kosovo between 2016 and 2022.

Food Category	No. of Tested Samples	No. of Positive Samples/No. of Tested Samples (%)	Ready-to-Eat Food	Food Products Consumed Cooked	Food Processing Environment	Raw Material
Meat and meat products	574	82/574 (14.29%)	20/286 (6.99%)	42/249 (16.87%)	-	20/39 (51.28%)
Milk and milk products	407	26/407 (6.39%)	21/361 (5.82%)	0/32 (0)	1/6 (16.66%)	4/8 (50.00%)
Fish and fishery products	9	4/9 (44.44%)	0/1 (0)	-	-	4/8 (50.00%)
Combined food products (meat, dairy)	5	5/5 (100.00%)	-	-	-	5/5 (100.00%)
Total	995	117/995 (11.76%)	41/648 (6.33%)	42/281 (14.95%)	1/6 (16.66%)	33/60 (55.00%)

**Table 2 foods-13-02883-t002:** The molecular characteristics of *L. monocytogenes* isolated from food products.

Food Chain	Food Category	Lineage	Molecular Serotype	Clonal Complex
Ready-to-eat food (n = 41)	Meat and meat products (20)	I (10)	IIb (1)	CC3 (1)
			IVb (9)	CC2 (1), CC4 (1), CC6 (6), CC315 (1)
		II (10)	IIa (8)	CC7 (1), CC8 (3), CC29 (1), CC37 (3)
			IIc (2)	CC9 (2)
	Milk and milk products (21)	I (10)	IIb (1)	CC87 (1)
			IVb (9)	CC2 (5), CC4 (2), ST32 (2)
		II (11)	IIa (10)	CC8 (1), CC14 (1), CC26 (1), CC29 (6), CC37 (1)
			IIc (1)	CC9 (1)
	Total	I (48.78%), II (51.22%)	IIb (2), IVb (18) IIa (18), IIc (3)	CC2 (6), CC3 (1), CC4 (3), CC6 (6), CC7 (1), CC8 (4), CC9 (3), CC 14 (1), CC26 (1), CC29 (7), ST32 (2), CC37 (4), CC87 (1), CC315 (1)
Food products consumed cooked (n = 42)	Meat and meat products (42)	I (10)	IIb (1)	CC5 (1)
			IVb (9)	CC2 (4), CC4 (1), CC6 (3), CC315 (1)
		II (32)	IIa (11)	CC7 (4), CC8 (1), CC14 (3), CC29 (1), CC37 (2)
			IIc (21)	CC9 (21)
	Total	I (23.81%), II (76.19%)	IIb (1), IVb (9) IIa (11), IIc (21)	CC2 (4), CC4 (1), CC5 (1), CC6 (3), CC7 (4), CC8 (1), CC9 (21), CC14 (3), CC29 (1), CC37 (2), CC315 (1)
Raw materials (n = 33)	Meat and meat products (20)	I (4)	IVb (4)	CC1 (4)
		II (16)	IIa (4)	CC7 (1), CC8 (3)
			IIc (12)	CC9 (12)
	Milk and milk products (4)	I (1)	IVb (1)	CC315 (1)
		II (3)	IIc (1)	CC9 (1)
			IIa (2)	CC8 (1), CC26 (1)
	Fish meat products (4)	I (3)	IIb (1)	CC87 (1)
			IVb (2)	CC2 (2)
		II (1)	IIa (1)	CC26 (1)
	Combined food products (5)	I (1)	IVb (1)	CC6 (1)
		II (4)	IIa (3)	CC8 (1), CC121 (2)
			IIc (1)	CC9 (1)
	Total	I (27.27%), II (72.73%)	IIb (1), IVb (8) IIa (10), IIc (14)	CC1 (4), CC2 (2), CC6 (1), CC7 (1), CC8 (5), CC9 (14), CC26 (2), CC87 (1), CC121 (2), CC315 (1)
Food contact sample (n = 1)	Environmental sample (1)	II (1)	IIa (1)	CC7 (1)
	Total	II (100.00%)	IIa (1)	CC7 (1)

## Data Availability

The original contributions presented in the study are included in the article, further inquiries can be directed to the corresponding author.

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
