# Peer review of "Molecular Characterization of Listeria monocytogenes in the Food Chain of the Republic of Kosovo from 2016 to 2022"

_foods, 2024, doi:10.3390/foods13182883_

Round 1
Reviewer 1 Report
Comments and Suggestions for Authors
This work aims to characterize Listeria monocytogenes strains (lineage, molecular serotype and CC) in order to provide an overview of the genetic diversity of the strains circulating among food products in Kosovo.
The paper is interesting and provides a detailed molecular characterization of the L. monocytogenes strains.
The manuscript is written quite clearly, the introduction provides a sufficient background and the methods are adequately described.
However, there are some parts that could be improved, in particular the abstract, the results and tha conclusion sections.
Abstract
-The abstract section should be revised, because some parts are difficult to understand. For example, the sentence “Most were isolated from meat and milk products” (line 26-27), what it refers to?
Results
-In my opinion, results should be better presented and described. In particular, in section 3.2. the percentages of the different serogroups and clonal complexes should be reported or in a table or in figure 2.
Conclusions
-The conclusions section is not present. It should be separated from the discussion section and better detailed.
Author Response
Dear Reviewer,
Please find below the response to your comments and suggestions
Comments 1: Abstract - The abstract section should be revised, because some parts are difficult to understand. For example, the sentence “Most were isolated from meat and milk products” (line 26-27), what it refers to?
Response 1:
We revised the abstract as follows:
“Abstract: The present study describes the genetic characterization of L. monocytogenes strains found in the Republic of Kosovo’s food chain. From 2016 to 2022, 995 samples were collected. Overall, 648 samples were from ready-to-eat (RTE) food products, 281 from food products consumed cooked (FPCC), 60 from raw materials, and six from environmental samples. Overall, 11.76% (117 out of 995) of samples were contaminated by L. monocytogenes, 6.33% (41 out of 648) RTE, 14.95% (42 out of 281) FPCC, 55.00% (33 out of 60) from raw materials, and 16.66% (1 out of 6) from environmental samples. All isolates were subjected to molecular serotyping and clonal complex (CCs) identification by using real-time PCR, as well as multilocus sequence typing. All isolates were grouped into four molecular serotypes, IIa (34.19%), IIb (3.48%), IIc (32.48%) and IVb (29.91%) as well as Lineage I (33.33%) and Lineage II (66.66%). Fourteen CCs were identified from 41 RTE isolates; however, CC29 (7), CC2 (6), and CC6 (6) were the most dominant. On the contrary, the CC9 is by far the most represented CC in both FPCC (21) and RM (14). Moreover, 30 isolates expressed CC1, CC2, CC4 or CC6 which are particularly associated with severe human infections.
Keywords: L. monocytogenes, food chain, phylogenetic lineage, molecular serotype, clonal complex.”
Comments 2: Results - In my opinion, results should be better presented and described. In particular, in section 3.2., the percentages of the different serogroups and clonal complexes should be reported or in a table or in figure 2.
Response 2:
We revised the text in the section 3.2 and additionally we included the table and figure in which lineages, molecular serotypes, and clonal complexes are described by categories:
Table 2: Molecular characteristics of L. monocytogenes isolated from food product
Food chain |
Food category |
Lineage |
Molecular serotype |
Clonal Complex |
Ready to eat food (n=41) |
Meat and meat product (20) |
I (10) |
IIb (1) |
CC3 (1) |
IVb (9) |
CC2 (1), CC4 (1), CC6 (6), CC315 (1) |
|||
II (10) |
IIa (8) |
CC7 (1), CC8 (3), CC29 (1), CC37 (3) |
||
IIc (2) |
CC9 (2) |
|||
Milk and milk product (21) |
I (10) |
IIb (1) |
CC87 (1) |
|
IVb (9) |
CC2 (5), CC4 (2), ST32 (2) |
|||
II (11) |
IIa (10) |
CC8 (1), CC14 (1), CC26 (1), CC29 (6), CC37 (1) |
||
IIc (1) |
CC9 (1) |
|||
Total |
I (48.78%), II (51.22%) |
IIb (2), IVb (18) IIa (18), IIc (3) |
CC2 (6), CC3 (1), CC4 (3), CC6 (6), CC7 (1), CC8 (4), CC9 (3), CC 14 (1), CC26 (1), CC29 (7), ST32 (2), CC37 (4), CC87 (1), CC315 (1) |
|
Food product consumed cooked (n=42) |
Meat and meat product (42) |
I (10) |
IIb (1) |
CC5 (1) |
IVb (9) |
CC2 (4), CC4 (1), CC6 (3), CC315 (1) |
|||
II (32) |
IIa (11) |
CC7 (4), CC8 (1), CC14 (3), CC29 (1), CC37 (2) |
||
IIc (21) |
CC9 (21) |
|||
Total |
I (23.81%), II (76.19% ) |
IIb (1), IVb (9) IIa (11), IIc (21) |
CC2 (4), CC4 (1), CC5 (1), CC6 (3), CC7 (4), CC8 (1), CC9 (21), CC14 (3), CC29 (1), CC37 (2), CC315 (1) |
|
Raw material (n=33) |
Meat and meat product (20) |
I (4) |
IVb (4) |
CC1 (4) |
II (16) |
IIa (4) |
CC7 (1), CC8 (3) |
||
IIc (12) |
CC9 (12) |
|||
Milk and milk product (4) |
I (1) |
IVb (1) |
CC315 (1) |
|
II (3) |
IIc (1) |
CC9 (1) |
||
IIa (2) |
CC8 (1), CC26 (1) |
|||
Fish meat product (4) |
I (3) |
IIb (1) |
CC87 (1) |
|
IVb (2) |
CC2 (2) |
|||
II (1) |
IIa (1) |
CC26 (1) |
||
Combined food products (5) |
I (1) |
IVb (1) |
CC6 (1) |
|
II (4) |
IIa (3) |
CC8 (1), CC121 (2) |
||
IIc (1) |
CC9 (1) |
|||
Total |
I (27.27%), II (72.73%) |
IIb (1), IVb (8) IIa (10), IIc (14) |
CC1 (4), CC2 (2), CC6 (1), CC7 (1), CC8 (5), CC9 (14), CC26 (2), CC87 (1), CC121 (2), CC315 (1) |
|
Food contact sample (n=1) |
Environmental sample (1) |
II (1) |
IIa (1) |
CC7 (1) |
Total |
II (100.00%) |
IIa (1) |
CC7 (1) |
Figure 2. Graphical distribution of L. monocytogenes clonal complexes in food chain
“3.2.2. Molecular serotypes
Among the 41 isolates of RTE, the molecular serotypes identified are 18 (43.90%) each of IIa and IVb, IIc 3 (7.32%), and IIb 2 (4.88%). From the 42 FPCC isolates, 21 (50.00%) belonged to IIc, followed by 11 (26.19%) of IIa, 9 (21.43) of IVb, and 1 (2.38%) of IIb molecular serotypes. From the 33 raw material isolates, 14 (42.42%) belonged to IIc, followed by 10 (30.30%) of IIa, 8 (24.24%) of IVb, and 1 (3.03%) of IIb molecular serotypes.
When divided by food category, out of 82 isolates from meat products only, IIc was the most represented molecular serotype, comprising 52 or 42.68% of isolates, followed by IIa 23 (28.05%), IVb 22 (22.26%), and IIb 2 (2.44%). Out of 25 isolates from milk and milk products alone, IIa was the most represented molecular serotype, comprising 12 (48%) of isolates, followed by IVb 10 (40.00%), IIc 2 (8.00%), and IIb 1 (4.00%) of isolates.
Moreover, out of 5 isolates from combined food isolates, 3 (60.00%) belonged to the IIa molecular serotype, followed by one (20.00%) IIc and one (20.00%) IVb. Two out of four (50.00%) isolates from fish products belonged to molecular serotypes IVb, IIa, and IIb, represented by one each. The isolate from the environmental samples belonged to molecular serotype IIa (Table 2).
3.3.3. Clonal complexes
Fourteen out of 17 clonal complexes were identified by real-time PCR from 41 RTE isolates; however, CC29, CC2, and CC6 were the most dominant, represented by 7 CC29, and followed by 6 CC2 and CC6 each clonal complexes. On the contrary, the CC9 by far is the most represented clonal complex in both FPCC (21) and RM (14). Thirteen groups of clonal complexes were identified from meat product only isolates. Thirty-five out of 82 meat product isolates expressed CC9, which is the most represented clonal complex in this group, followed by CC6 expressed by 9, CC8 by 7, and CC2 expressed by 5 isolates. Eleven groups of clonal complexes were identified from 25 milk products only. CC29 was represented by 6 isolates, followed by CC2 by 5, while CC9 was represented by only 2 isolates. Three of the 4 isolates from fish products belonged to CC2 with 2, and CC26, CC87 with only one each. From 4 groups of clonal complexes isolated from combined food products, we found that CC121, which we have not encountered in any other food category was expressed by 2 isolates, whereas CC6, CC8 and CC9 were expressed by one isolate each (Table 2 & Figure 2).”
Comments 3: Conclusions - The conclusions section is not present. It should be separated from the discussion section and better detailed.
Response 3:
The conclusion is presented in point 5:
“Conclusion: As a conclusion of this study covering the period 2016-2022, which included samples from the Republic of Kosovo, we obtained results related to the general condition of L. monocytogenes. We showed that contamination in RTE and meat products consumed cooked is higher than the prevalence previously reported in the EU. Major hypervirulent clones CC1, CC2, CC4, and CC6 were identified in food products. These clones should be targeted as a priority using more rigorous control measures, especially to protect at-risk groups from this pathogen (the elderly population, pregnant women, children, and immunosuppressed persons). Effective management of the risk of L. monocytogenes in the food chain and in food products requires a thorough assessment of sources of contamination, continuous monitoring and rigorous control measures. By conducting detailed risk assessments and adapting strategies based on ongoing assessments, it is possible to minimize or completely eliminate the risk of listeriosis and protect public health. It is necessary to continue active monitoring of the entire food chain to support real-time molecular surveillance.
- Facilitate outbreak investigation and surveillance by pre-screening strains.
- Identify processing plants with hypervirulent clones and monitor them in real time with adequate methods.
This study might be extended to the food processing environment, which is often a major source of contamination. Within the food processing environment, better monitoring will identify the major source of contamination, leading to improved management and a reduction of contamination, in particular by hypervirulent strains.”
Reviewer 2 Report
Comments and Suggestions for Authors
Abstract: The study determines the serotypes, but they are not mentioned in the current abstract. To consider them as hypervirulent clones (L35), it is necessary to review at least the relevant virulence factors in silico.
Introduction: Lines 93-94: It is important to review the information on MLST, as the gold standard for outbreak investigation is whole genome sequencing (WGS).
Lines 94-95: It is recommended to expand on the relevance of identifying clonal complexes (CCs) using PCR.
Materials and Methods: Lines 103-141: It is suggested to separate the sections for sampling, enumeration, isolation, and biochemical identification.
Results: Lines 201-208: It is recommended to create a table with information on serotypes, CCs, and lineages.
Line 209: The MST appears to refer to MLST and CCs of the strains, which is confusing. It is important to distinguish between ST and CC separately. Additionally, it is mentioned that each area contains the same genoserotype, but the specific types are not detailed. These details need to be specified.
Discussion: The discussion is very brief, likely because the generated information is insufficient for a comprehensive and relevant manuscript. The study aims to characterize L. monocytogenes, and while genotype information is provided, relevant details on virulence and resistance factors are missing. For example, there are methodologies related to serotype determination, but nothing is included in this section. Therefore, this part needs to be rewritten to align with the study's objective and emphasize its relevance in terms of food production and listeriosis prevention, as mentioned in the introduction (lines 47-55).
Author Response
Dear Reviewer,
Please find below the response to your comments and suggestions
Comments 1: Abstract: The study determines the serotypes, but they are not mentioned in the current abstract.
Response 2: We revised the abstract as follows:
“Abstract: The present study describes the genetic characterization of L. monocytogenes strains found in the Republic of Kosovo’s food chain. From 2016 to 2022, 995 samples were collected. Overall, 648 samples were from ready-to-eat (RTE) food products, 281 from food products consumed cooked (FPCC), 60 from raw materials, and six from environmental samples. Overall, 11.76% (117 out of 995) of samples were contaminated by L. monocytogenes, 6.33% (41 out of 648) RTE, 14.95% (42 out of 281) FPCC, 55.00% (33 out of 60) from raw materials, and 16.66% (1 out of 6) from environmental samples. All isolates were subjected to molecular serotyping and clonal complex (CCs) identification by using real-time PCR, as well as multilocus sequence typing. All isolates were grouped into four molecular serotypes, IIa (34.19%), IIb (3.48%), IIc (32.48%) and IVb (29.91%) as well as Lineage I (33.33%) and Lineage II (66.66%). Fourteen CCs were identified from 41 RTE isolates; however, CC29 (7), CC2 (6), and CC6 (6) were the most dominant. On the contrary, the CC9 is by far the most represented CC in both FPCC (21) and RM (14). Moreover, 30 isolates expressed CC1, CC2, CC4 or CC6 which are particularly associated with severe human infections.
Keywords: L. monocytogenes, food chain, phylogenetic lineage, molecular serotype, clonal complex.”
Comments 2: To consider them as hypervirulent clones (L35), it is necessary to review at least the relevant virulence factors in silico.
Response 2: We rephrased the sentence in the abstract:
“Moreover, 30 isolates expressed CC1, CC2, CC4 or CC6 which are particularly associated with severe human infections.”
We additionally elaborated in the discussion part:
“Moreover, in the present study, 30 isolates expressed CC1, CC2, CC4, or CC6, which are particularly associated with severe human infections [28,35]. This highlights the need to map the distribution of these CCs in the current food chain in order to predict the risk for public health associated with food products.”
Comments 3: Introduction: Lines 93-94: It is important to review the information on MLST, as the gold standard for outbreak investigation is whole genome sequencing (WGS). Lines 94-95: It is recommended to expand on the relevance of identifying clonal complexes (CCs) using PCR.
Response 3: In the introduction part, information related to MLST and WGS is presented:
“New techniques such as typing based on whole genome sequencing (WGS) are used in European countries as a highly accurate "Gold standard" tool in outbreak research and tracing the source of listeriosis [13, 26]. Also, the MLST method, which is used as a tool for outbreak investigation and genetic mapping studies of L. monocytogenes populations [26]. Since these methods are expensive and require a longer time to reach the result, today efforts are made for fast techniques by means of multiplex PCR. Recently, a multiplex PCR method has been developed for the rapid identification of 30 CC of L. monocytogenes [27].
- Pietzka, A.; Allerberger, F.; Murer, A.; Lennkh, A.; Stoger, A.; Rosel, A.C.; Huhulescu, S.; Maritschnik, S.; Springer, B.; Lepuschitz, S. Whole genome sequencing based surveillance of L. monocytogenes for early detection and investigations of listeriosis outbreaks, Front. Public Health 7, 2019, 139, https://doi.org/ 10.3389/fpubh.2019.00139.”
Comments 4: Materials and Methods: Lines 103-141: It is suggested to separate the sections for sampling, enumeration, isolation, and biochemical identification.
Response 4: It is divided into three sections:
“2.1. Sampling
2.2. Detection and isolation
2.3. Biochemical identification
Also the word "counting" is omitted altogether since the method is about "detection".
Comments 5: Results: Lines 201-208: It is recommended to create a table with information on serotypes, CCs, and lineages.
Response 5: We revised the text in the section 3.2 (requested by reviewer 1) and additionally we included the table in which lineages, molecular serotypes, and clonal complexes are described by categories:
Table 2: Molecular characteristics of L. monocytogenes isolated from food product
Food chain |
Food category |
Lineage |
Molecular serotype |
Clonal Complex |
Ready to eat food (n=41) |
Meat and meat product (20) |
I (10) |
IIb (1) |
CC3 (1) |
IVb (9) |
CC2 (1), CC4 (1), CC6 (6), CC315 (1) |
|||
II (10) |
IIa (8) |
CC7 (1), CC8 (3), CC29 (1), CC37 (3) |
||
IIc (2) |
CC9 (2) |
|||
Milk and milk product (21) |
I (10) |
IIb (1) |
CC87 (1) |
|
IVb (9) |
CC2 (5), CC4 (2), ST32 (2) |
|||
II (11) |
IIa (10) |
CC8 (1), CC14 (1), CC26 (1), CC29 (6), CC37 (1) |
||
IIc (1) |
CC9 (1) |
|||
Total |
I (48.78%), II (51.22%) |
IIb (2), IVb (18) IIa (18), IIc (3) |
CC2 (6), CC3 (1), CC4 (3), CC6 (6), CC7 (1), CC8 (4), CC9 (3), CC 14 (1), CC26 (1), CC29 (7), ST32 (2), CC37 (4), CC87 (1), CC315 (1) |
|
Food product consumed cooked (n=42) |
Meat and meat product (42) |
I (10) |
IIb (1) |
CC5 (1) |
IVb (9) |
CC2 (4), CC4 (1), CC6 (3), CC315 (1) |
|||
II (32) |
IIa (11) |
CC7 (4), CC8 (1), CC14 (3), CC29 (1), CC37 (2) |
||
IIc (21) |
CC9 (21) |
|||
Total |
I (23.81%), II (76.19% ) |
IIb (1), IVb (9) IIa (11), IIc (21) |
CC2 (4), CC4 (1), CC5 (1), CC6 (3), CC7 (4), CC8 (1), CC9 (21), CC14 (3), CC29 (1), CC37 (2), CC315 (1) |
|
Raw material (n=33) |
Meat and meat product (20) |
I (4) |
IVb (4) |
CC1 (4) |
II (16) |
IIa (4) |
CC7 (1), CC8 (3) |
||
IIc (12) |
CC9 (12) |
|||
Milk and milk product (4) |
I (1) |
IVb (1) |
CC315 (1) |
|
II (3) |
IIc (1) |
CC9 (1) |
||
IIa (2) |
CC8 (1), CC26 (1) |
|||
Fish meat product (4) |
I (3) |
IIb (1) |
CC87 (1) |
|
IVb (2) |
CC2 (2) |
|||
II (1) |
IIa (1) |
CC26 (1) |
||
Combined food products (5) |
I (1) |
IVb (1) |
CC6 (1) |
|
II (4) |
IIa (3) |
CC8 (1), CC121 (2) |
||
IIc (1) |
CC9 (1) |
|||
Total |
I (27.27%), II (72.73%) |
IIb (1), IVb (8) IIa (10), IIc (14) |
CC1 (4), CC2 (2), CC6 (1), CC7 (1), CC8 (5), CC9 (14), CC26 (2), CC87 (1), CC121 (2), CC315 (1) |
|
Food contact sample (n=1) |
Environmental sample (1) |
II (1) |
IIa (1) |
CC7 (1) |
Total |
II (100.00%) |
IIa (1) |
CC7 (1) |
Figure 2. Graphical distribution of L. monocytogenes clonal complexes in food chain
“3.2.2. Molecular serotypes
Among the 41 isolates of RTE, the molecular serotypes identified are 18 (43.90%) each of IIa and IVb, IIc 3 (7.32%), and IIb 2 (4.88%). From the 42 FPCC isolates, 21 (50.00%) belonged to IIc, followed by 11 (26.19%) of IIa, 9 (21.43) of IVb, and 1 (2.38%) of IIb molecular serotypes. From the 33 raw material isolates, 14 (42.42%) belonged to IIc, followed by 10 (30.30%) of IIa, 8 (24.24%) of IVb, and 1 (3.03%) of IIb molecular serotypes.
When divided by food category, out of 82 isolates from meat products only, IIc was the most represented molecular serotype, comprising 52 or 42.68% of isolates, followed by IIa 23 (28.05%), IVb 22 (22.26%), and IIb 2 (2.44%). Out of 25 isolates from milk and milk products alone, IIa was the most represented molecular serotype, comprising 12 (48%) of isolates, followed by IVb 10 (40.00%), IIc 2 (8.00%), and IIb 1 (4.00%) of isolates.
Moreover, out of 5 isolates from combined food isolates, 3 (60.00%) belonged to the IIa molecular serotype, followed by one (20.00%) IIc and one (20.00%) IVb. Two out of four (50.00%) isolates from fish products belonged to molecular serotypes IVb, IIa, and IIb, represented by one each. The isolate from the environmental samples belonged to molecular serotype IIa (Table 2).
3.3.3. Clonal complexes
Fourteen out of 17 clonal complexes were identified by real-time PCR from 41 RTE isolates; however, CC29, CC2, and CC6 were the most dominant, represented by 7 CC29, and followed by 6 CC2 and CC6 each clonal complexes. On the contrary, the CC9 by far is the most represented clonal complex in both FPCC (21) and RM (14). Thirteen groups of clonal complexes were identified from meat product only isolates. Thirty-five out of 82 meat product isolates expressed CC9, which is the most represented clonal complex in this group, followed by CC6 expressed by 9, CC8 by 7, and CC2 expressed by 5 isolates. Eleven groups of clonal complexes were identified from 25 milk products only. CC29 was represented by 6 isolates, followed by CC2 by 5, while CC9 was represented by only 2 isolates. Three of the 4 isolates from fish products belonged to CC2 with 2, and CC26, CC87 with only one each. From 4 groups of clonal complexes isolated from combined food products, we found that CC121, which we have not encountered in any other food category was expressed by 2 isolates, whereas CC6, CC8 and CC9 were expressed by one isolate each (Table 2 & Figure 2).”
Comments: Line 209: The MST appears to refer to MLST and CCs of the strains, which is confusing. It is important to distinguish between ST and CC separately. Additionally, it is mentioned that each area contains the same genoserotype, but the specific types are not detailed. These details need to be specified.
Response 6: The figure 2 legend was modified to be clearer on MST building. The Genoserogroup (so called molecular serotype) were highlighted in the figure, and specified in the figure legend:
Figure 3. (a) Multi locus sequence typing (MLST) clonal complexes (CCs) minimum spanning tree (MST) of the 117 L. monocytogenes strains of the study panel. Each CC is indicated by a circular node whose size reflects the number of strains. The CCs used to build the MST were obtained by an MLST alternative method, which provide only the CC, to build the MST the smallest ST allelic code within the CC was used. The numbers along the node connecting lines indicate the number of allelic differences between them The colour reflects the food category: meat products in red, milk products in yellow, fish products in blue, and combined food products in green. Each delimited area groups the CCs belonging to the same molecular serotype, indicated in a black frame (b) Strain distribution according to lineage and type of food sector.
Comments 7: Discussion: The discussion is very brief, likely because the generated information is insufficient for a comprehensive and relevant manuscript. The study aims to characterize L. monocytogenes, and while genotype information is provided, relevant details on virulence and resistance factors are missing. For example, there are methodologies related to serotype determination, but nothing is included in this section. Therefore, this part needs to be rewritten to align with the study's objective and emphasize its relevance in terms of food production and listeriosis prevention, as mentioned in the introduction (lines 47-55).
Response 7: Sentences have been added and changed including:
The highest contamination rates were found in raw materials (55.00%) and FPCC (14.95%), whereas RTE foods had the lowest prevalence (6.33%). This distribution high-lights the risk associated with raw and minimally processed food products, which are more susceptible to contamination due to factors such as handling and storage conditions.
The results of molecular serotyping provided a pre-screening analysis prior CC iden-tification. Molecular serotype results enables to connect with former studies using only conventional serotyping or molecular serotyping as typing method. Both methods provide a corresponding nomenclature, in short, with few exceptions, major serotype correspond-ence is 1/2a=IIa, 1/2b=IIb, 1/2c=IIc and 4b=IVb. In present study four main molecular sero-types were identified: IIa (34.19%), IIb (3.48%), IIc (32.48%), and IVb (29.91%). The pre-dominance of molecular serotype IIa and IIc, accounting for over two thirds of the isolates, aligns with previous studies reporting these molecular serotypes as prevalent in various geographical regions [38,39,40]. The diversity of molecular serotypes is indicative of the varied sources and potential reservoirs of contamination within the food chain. Regarding IIc (or 1/2c) strains in meat products, they are largely reported in former studies [41,42,43] conversely, they are rare in milk products [44,45]. These results show the large distribution IIc (or 1/2c) strain in Europe during a long period of time, including RKS. Regarding IVb (or 4b) strains, this molecular serotype includes CCs associated with severe human infections [35] (CC1, CC2, CC4 and CC6) and considered as hypervirulent. Overall, 26.83% of the meat and 40.00% of the milk product strains belonged to this molecular serotype.
Overall, 14 CCs were identified among the 41 RTE isolates, with CC29, CC2, and CC6 being the most dominant. In contrary, CC9 emerged as the most prevalent in FPCC and raw materials. This clonal complex is known for its presence in various environmental and food sources but is less commonly associated with severe human cases [46]. The high prevalence of CC9 in raw materials could suggest that it is a persistent environmental strain, possibly contributing to cross-contamination during food processing. Surprisingly, CC121 was absent from meat products in the present study, although it was frequently associated with meat products elsewhere [35]. The absence of this clone in meat products in the current study needs further investigation but may reflect a difference in animal slaughtering or processing practices in Kosovo.
All of 30 isolates expressing CCs associated with severe human infections (CC1, CC2, CC4, or CC6), belonged to molecular serotype IVb. This highlights the need for identify and confirm the presence of these CCs and map their distribution in the current food chain in order to predict the risk for public health associated with food products.
Whole-genome sequencing (WGS) is recommended to be used in the future in or-der to provide additional information on virulence factors and other genetic determinants.
Comments 7: No conclusions present.
Response 7: According to your recommendation, the Conclusion has also been added:
“5. Conclusion: As a conclusion of this study covering the period 2016-2022, which included samples from the Republic of Kosovo, we obtained results related to the general condition of L. monocytogenes. We showed that contamination in RTE and meat products consumed cooked is higher than the prevalence previously reported in the EU. Major hypervirulent clones CC1, CC2, CC4, and CC6 were identified in food products. These clones should be targeted as a priority using more rigorous control measures, especially to protect at-risk groups from this pathogen (the elderly population, pregnant women, children, and immunosuppressed persons). Effective management of the risk of L. monocytogenes in the food chain and in food products requires a thorough assessment of sources of contamination, continuous monitoring and rigorous control measures. By conducting detailed risk assessments and adapting strategies based on ongoing assessments, it is possible to minimize or completely eliminate the risk of listeriosis and protect public health. It is necessary to continue active monitoring of the entire food chain to support real-time molecular surveillance.
- Facilitate outbreak investigation and surveillance by pre-screening strains.
- Identify processing plants with hypervirulent clones and monitor them in real time with adequate methods.
This study might be extended to the food processing environment, which is often a major source of contamination. Within the food processing environment, better monitoring will identify the major source of contamination, leading to improved management and a reduction of contamination, in particular by hypervirulent strains.”
Added References:
- Korsak D, Borek A, Daniluk S, Grabowska A, Pappelbaum K. Antimicrobial susceptibilities of Listeria monocytogenes strains isolated from food and food processing environment in Poland. Int J Food Microbiol. 2012, 158(3):203-208. doi:10.1016/j.ijfoodmicro.2012.07.016.
- Maćkiw, E.; Stasiak, M.; Kowalska, J.; Kucharek, K.; Korsak, D.; Postupolski, J. Occurrence and Characteristics of Listeria monocytogenes in Ready-to-Eat Meat Products in Poland. Journal of Food Protection, 2020, 83, 6, 1002-1009, https://doi.org/10.4315/JFP-19-525.
- Daza, P.B.; Pietzka, A.; Martinovic, A.; Ruppitsch, W.; Zuber, B.I. Surveillance and genetic characterization of Listeria monocytogenes in the food chain in Montenegro during the period 2014–2022. Front. Microbiol., 2024, 15:1418333. doi: 10.3389/fmicb.2024.1418333.
- Thévenot, D., et al. "Fate of Listeria monocytogenes in experimentally contaminated French sausages." International Journal of Food Microbiology 101.2, 2005, 189-200, https://doi.org/10.1016/j.ijfoodmicro.2004.11.006.
42.Meloni, D.; Piras, F.; Mureddu, A.; Fois, F.; Consolati, S.G.; Lamon, S.; Mazzette, R. Listeria monocytogenes in five Sardinian swine slaughterhouses: Prevalence, Serotype and Genotype Characterization. J. Food Prot. 2013, 76, 1863–1867, https://doi.org/10.4315/0362-028X.JFP-12-505.
- Kramarenko, Toomas, et al. "Listeria monocytogenes prevalence and serotype diversity in various foods." Food control 30.1 2013, 24-29, https://doi.org/10.1016/j.foodcont.2012.06.047.
44.Wagner, M., et al. "Characterization of Listeria monocytogenes isolates from 50 small-scale Austrian cheese factories." Journal of food protection 69.6, 2006, 1297-1303, https://doi.org/10.4315/0362-028X-69.6.1297.
45.Kiss, Réka, et al. "Listeria monocytogenes food monitoring data and incidence of human listeriosis in Hungary, 2004." International journal of food microbiology 112.1, 2006, 71-74, https://doi.org/10.1016/j.ijfoodmicro.2006.06.013.
- Maury, M.M.; Bracq-Dieye, H.; Huang, L. et al.Hypervirulent Listeria monocytogenesclones’ adaption to mammalian gut accounts for their association with dairy products. Nat Commun 10, 2488, 2019. https://doi.org/10.1038/s41467-019-10380-0.
Reviewer 3 Report
Comments and Suggestions for Authors
This paper conducts molecular surveillance research on Listeria monocytogenes in Kosovo over a certain period, with a large sample size (995) and employs the classic MLST analysis technique for an in-depth study. The overall detection results are higher than those reported in existing studies, and the reasons for this need to be explained and interpreted, especially regarding the sampling basis and methods in Kosovo. Detailed issues include the following: (1) Why is there only one fish sample? (2) There is a lack of a method description for data processing and statistical analysis, including the names of software applications, version numbers, and developing companies. (3) The discussion and analysis in this paper are too simplistic and lack deeper explanations, such as the relationship between CC and ST, and the analysis of typing and pathogenicity. It is recommended to supplement the relevant content from the FAO/WHO MRA38 report. (4) Simple detection and analysis are far from enough. Comparisons and deep analysis of possible causes are also needed. Risk monitoring prepares for risk assessment and prevention and control. So, how should risk assessment and management control be carried out subsequently? This is the most meaningful point for extension. It is recommended to further refine and elevate this aspect and include it in the final conclusions or outlook.
Author Response
Dear Reviewer,
Please find below the response to your comments and suggestions
This paper conducts molecular surveillance research on Listeria monocytogenes in Kosovo over a certain period, with a large sample size (995) and employs the classic MLST analysis technique for an in-depth study. The overall detection results are higher than those reported in existing studies, and the reasons for this need to be explained and interpreted, especially regarding the sampling basis and methods in Kosovo.
Detailed issues include the following:
Comments 1: Why is there only one fish sample?
Response 1: In fact, during the period under investigation, we had a total of 10 fish samples, of which four turned out positive for L. monocytogenes. As all samples received in our laboratory were part of the sampling framework from the third party, we had no influence on it.
Comments 2: There is a lack of a method description for data processing and statistical analysis, including the names of software applications, version numbers, and developing companies.
Response 2: “The strain CC obtained was displayed within a minimum spanning tree, built using BioNumerics 7.6.3 (BioMérieux Applied Maths, Sint-Martens-Latem, Belgium), with default parameters.”
Comments 3: The discussion and analysis in this paper are too simplistic and lack deeper explanations, such as the relationship between CC and ST, and the analysis of typing and pathogenicity. It is recommended to supplement the relevant content from the FAO/WHO MRA38 report.
Response 3: We added the following sentence in the section Materials and Methods:
“Additionally, multi locus sequence typing (MLST) was used to complement the results with ST32, which was not covered by the CCs real-time PCR panel.”
(The figure 3 legend was modified to be clearer on MST building. The Genoserogroup (so called molecular serotype) were highlighted in the figure, and specified in the figure legend).
“Figure 3. (a) Multi locus sequence typing (MLST) clonal complexes (CCs) minimum spanning tree (MST) of the 117 L. monocytogenes strains of the study panel. Each CC is indicated by a circular node whose size reflects the number of strains. The CCs used to build the MST were obtained by an MLST alternative method, which provide only the CC, to build the MST the smallest ST allelic code within the CC was used. The numbers along the node connecting lines indicate the number of allelic differences between them The colour reflects the food category: meat products in red, milk products in yellow, fish products in blue, and combined food products in green. Each delimited area groups the CCs belonging to the same molecular serotype, indicated in a black frame (b) Strain distribution according to lineage and type of food sector.”
Comments 4: Simple detection and analysis are far from enough. Comparisons and deep analysis of possible causes are also needed. Risk monitoring prepares for risk assessment and prevention and control. So, how should risk assessment and management control be carried out subsequently? This is the most meaningful point for extension. It is recommended to further refine and elevate this aspect and include it in the final conclusions or outlook.
Response 4: The conclusion is presented in point 5:
“Conclusion: As a conclusion of this study covering the period 2016-2022, which included samples from the Republic of Kosovo, we obtained results related to the general condition of L. monocytogenes. We showed that contamination in RTE and meat products consumed cooked is higher than the prevalence previously reported in the EU. Major hypervirulent clones CC1, CC2, CC4, and CC6 were identified in food products. These clones should be targeted as a priority using more rigorous control measures, especially to protect at-risk groups from this pathogen (the elderly population, pregnant women, children, and immunosuppressed persons). Effective management of the risk of L. monocytogenes in the food chain and in food products requires a thorough assessment of sources of contamination, continuous monitoring and rigorous control measures. By conducting detailed risk assessments and adapting strategies based on ongoing assessments, it is possible to minimize or completely eliminate the risk of listeriosis and protect public health. It is necessary to continue active monitoring of the entire food chain to support real-time molecular surveillance.
- Facilitate outbreak investigation and surveillance by pre-screening strains.
- Identify processing plants with hypervirulent clones and monitor them in real time with adequate methods.
This study might be extended to the food processing environment, which is often a major source of contamination. Within the food processing environment, better monitoring will identify the major source of contamination, leading to improved management and a reduction of contamination, in particular by hypervirulent strains.”
Round 2
Reviewer 1 Report
Comments and Suggestions for Authors
The manuscript has been improved. The abstract section is clear ad well written and results are clearly described.
Author Response
Comments 1: The manuscript has been improved. The abstract section is clear ad well written and results are clearly described.
Response 1: Thank you for your comments and expert review. Your work to help us improve the manuscript is greatly appreciated.
Reviewer 2 Report
Comments and Suggestions for Authors
Aquí tienes la traducción al inglés del párrafo mejorado:
Lines 94-96: What the authors propose does not align with the cited articles. The method used is the core genome MLST (cgMLST), gene by gene, derived from whole genome sequencing (WGS) for outbreak investigations, as clearly explained in reference No. 26 (Pietzka et al., 2024). Therefore, the authors should correct this error.
Lines 96-99: This statement is also incorrect. WGS, using technologies like Nanopore, is fast and can be done in real-time. Additionally, the use of clonal complexes (CCs) has the limitation of identifying only one clonal complex, which reduces the amount of information obtained. For example, in Listeria monocytogenes, ST2349 and ST5 belong to CC5; however, only ST5 is of clinical importance. The same situation occurs with CC2, a clone considered hypervirulent. Therefore, the method used is only useful as a screening tool for CCs. The paragraph should be rewritten to avoid misleading readers.
Lines 375-378: The authors need to be more critical of the new method, as it has limitations that are not mentioned here. It is suggested to avoid using the phrase "completely eliminate the risk," as risk is always greater than 0.
Lines 380-386: This section is redundant and ambiguous. It is recommended to rewrite or remove it, as the conclusion is already mentioned in lines 377-379 and can be supplemented with the information in lines 370-373.
Author Response
Dear Reviewer, thank you for your expertise and the very helpful comments and suggestions. A detailed response is provided below:
Comments 1:
Lines 94-96: What the authors propose does not align with the cited articles. The method used is the core genome MLST (cgMLST), gene by gene, derived from whole genome sequencing (WGS) for outbreak investigations, as clearly explained in reference No. 26 (Pietzka et al., 2024). Therefore, the authors should correct this error.
Lines 96-99: This statement is also incorrect. WGS, using technologies like Nanopore, is fast and can be done in real-time. Additionally, the use of clonal complexes (CCs) has the limitation of identifying only one clonal complex, which reduces the amount of information obtained. For example, in Listeria monocytogenes, ST2349 and ST5 belong to CC5; however, only ST5 is of clinical importance. The same situation occurs with CC2, a clone considered hypervirulent. Therefore, the method used is only useful as a screening tool for CCs. The paragraph should be rewritten to avoid misleading readers.
Response 1:
“Today, with access to advanced technology such as whole genome sequencing (WGS) and assessment of the core genome multilocus sequence typing MLST (cgMLST) from the data derived from WGS, it is possible to type and discriminate with high accuracy different strains of outbreak, thus tracing the source of listeriosis [13, 26, 27]. However, not all laboratories in developing countries have access to WGS services in a reasonable time during outbreaks. Therefore, an approach targeting CCs by high-throughput real-time PCR, which enables the rapid identification of the 30 major MLST CCs circulating in Europe, is particularly helpful for strain screening prior to WGS [28].”
Comments 2:
Lines 375-378: The authors need to be more critical of the new method, as it has limitations that are not mentioned here. It is suggested to avoid using the phrase "completely eliminate the risk," as risk is always greater than 0.
Lines 380-386: This section is redundant and ambiguous. It is recommended to rewrite or remove it, as the conclusion is already mentioned in lines 377-379 and can be supplemented with the information in lines 370-373.
Response 2:
For lines 375-378, we changed and rephrased the sentences and incorporated them in the same paragraph with the responses for the comment on lines 380-386, removing repeated content as suggested.
“By conducting detailed risk assessments and adapting strategies based on ongoing assessments, it is possible to reduce the risk of listeriosis and protect public health. It is necessary to continue active monitoring of the entire food chain to support in real-time molecular surveillance of CCs present, and as soon as possible to investigate at least retrospectively with WGS and cgMLST.”